# Scutellarin Alleviates Ischemic Brain Injury in the Acute Phase by Affecting the Activity of Neurotransmitters in Neurons

**DOI:** 10.3390/molecules28073181

**Published:** 2023-04-04

**Authors:** Chunguo Wang, Yaoyu Liu, Xi Liu, Yuting Zhang, Xingli Yan, Xinqi Deng, Jinli Shi

**Affiliations:** 1School of Chinese Materia Medica, Beijing University of Chinese Medicine, Beijing 100105, China; 2Beijing Research Institute of Chinese Medicine, Beijing University of Chinese Medicine, Beijing 100105, China; 3School of Trational Chinese Medicine, Beijing University of Chinese Medicine, Beijing 100105, China; 4Institute of Chinese Materia Medica, China Academy of Chinese Medical Sciences, Beijing 100091, China

**Keywords:** cerebral ischemic stroke, neurotransmitter, scutellarin, acute phase, mechanism

## Abstract

Cerebral ischemic stroke is a common neuron loss disease that is caused by the interruption of the blood supply to the brain. In order to enhance the CIS outcome, both identifying the treatment target of ischemic brain damage in the acute phase and developing effective therapies are urgently needed. Scutellarin had been found to be beneficial to ischemic injuries and has been shown to have potent effects in clinical application on both stroke and myocardial infarction. However, whether scutellarin improves ischemic brain damage in the acute phase remains unknown. In this study, the protective effects of scutellarin on ischemic brain damage in the acute phase (within 12 h) were illustrated. In middle cerebral artery occlusion and reperfusion (MCAO/R) modeling rats, the Z-Longa score was significantly down-regulated by 25% and 23.1%, and the brain infarct size was reduced by 26.95 ± 0.03% and 25.63 ± 0.02% when responding to high-dose and low-dose scutellarin treatments, respectively. H&E and TUNEL staining results indicated that the neuron loss of the ischemic region was improved under scutellarin treatment. In order to investigate the mechanism of scutellarin’s effects on ischemic brain damage in the acute phase, changes in proteins and metabolites were analyzed. The suppression of scutellarin on the glutamate-inducing excitatory amino acid toxicity was strongly indicated in the study of both proteomics and metabolomics. A molecular docking experiment presented strong interactions between scutellarin and glutamate receptors, which score much higher than those of memantine. Further, by performing a parallel reaction monitoring-mass spectrometry (PRM-MS) study on both the cortex and hippocampus tissue of the ischemic region, we screened the scutellarin-regulating molecules that are involved in both the release and transportation of neurotransmitters. It was found that the aberrant levels of glutamate receptors, including EAAT2, GRIN1, GRIN2B, and GRM1, as well as of other glutamatergic pathway-involving proteins, including CAMKK2, PSD95, and nNOS, were significantly regulated in the ischemic cortex. In the hippocampus, EAAT2, GRIN1, nNOS, and CAM were significantly regulated. Taken together, scutellarin exerts potent effects on ischemic brain damage in the acute phase by regulating the activity of neurotransmitters and reducing the toxicity of excitatory amino acids in in neurons.

## 1. Introduction

With a high mortality and morbidity rate, and the potential for producing long-term disabilities, cerebral ischemic stroke has attracted much attention in the medical field. Intervention during the acute phase is crucial in the treatment of strokes [1].

Intravenous thrombolysis and thrombectomy have long been considered as the first-line treatments for cerebral ischemic stroke [2,3]. Due to the narrow therapeutic window for thrombolytics, the progressing course of a stroke during the acute phase, as well as the lack of fast-acting drugs that can prevent neuron damage, most stroke survivors suffer from a more severe prognosis [4,5]. The development and application of MCAO/R animal models have improved our understanding of the mechanisms and potential targets of ischemic stroke [6]. Although knowledge about the acute phase of ischemic strokes has advanced, drug development in this field remains disappointing, since the tested drugs often possess high toxicity or low efficacy [7,8,9]. Therefore, current therapeutic approaches are far from fulfilling social and personal satisfaction in terms of individuals’ quality of life [3,10]. Therefore, enhanced knowledge about the molecules that are available for use in order to rapidly intervene in an ischemic stroke is urgently needed.

Seeking references to the effective remedy of diseases in natural medicine has become appealing in terms of providing new perspectives on the development of drugs. Scutellarin, also known as scutellarein 7-glucuronide, is a flavonoid that exists in multiple plants, including Arabidopsis thaliana [11], Perilla frutescens [12,13], Scutellaria baicalensis [14], Scutellaria polyodon [15], and Erigeron breviscapus. In previous studies, Scutellarin has been found to be beneficial to ischemic diseases, such as stroke and myocardial infarction, by fighting against inflammation, oxidation, and coagulation. It shows high efficacy and low toxicity in clinical applications [16]. However, evidence of scutellarin’s effect on acute ischemic brain damage remains insufficient. Acute cerebral ischemic stroke is caused by the interruption of cerebral circulation, which leads to rapid failures in neuronal electrical activity, energy state, and ion homeostasis [17,18,19]. In this study, we demonstrate the potent alleviating effects of scutellarin on neuron apoptosis and the loss of neurological function within 12 h after MCAO/R modeling. The results of Z-Longa scoring, TTC, H&E, and TUNEL staining, as well as Western blotting, illustrate how scutellarin acts quickly to generate neuroprotection. Moreover, the mechanisms of scutellarin treatment during acute cerebral ischemia are investigated in the cortex and hippocampus areas, respectively, from the perspective of the expression level of proteins and metabolites, as well as of drug–target interactions. We also suggest that scutellarin exerts quick regulatory effects on neurotransmitter activities, especially in the glutamate-related pathway. This study aims to provide evidence regarding scutellarin’s protection of neurons after acute ischemic injury, and improve knowledge regarding the therapeutic molecules that target the activity of neuro-transmitters in neurons.

## 2. Results

### 2.1. Scutellarin Protects Neurons against Ischemic Injury in Acute Phase

The Z-Longa scoring results showed that the impairment of nerve function could be significantly improved in the acute phase (12 h after reperfusion) (Figure 1B) when treated with 98% pure scutellarin (Figure 1A); this suggests scutellarin’s efficacy when treating acute ischemic brain injury. The brain infarction was found to increase by 33.55 ± 0.03% at 12 h after reperfusion. Under low-dose and high-dose scutellarin administration, the brain infarction size was decreased to 26.95 ± 0.03% and 25.63 ± 0.02%, respectively (Figure 1C,D). In terms of substructures, neuronal damage in the cortex and hippocampus, including nucleus pyknosis, chromatin condensation, enlarged intercellular space, and cavities formed by dissolved cells, was improved upon scutellarin administration at both low and high doses (Figure 1E). In addition, compared to the model group, the number of dead cells (Figure 1F,G) were significantly reduced by 51.1% and 80.9% in the ischemic region upon the administration of low and high doses of scutellarin, respectively. The expression of cleaved caspase 3 was also suppressed (Figure 1H,I). Taken together, scutellarin shows that it is able to potently protect ischemic neurons from apoptosis in the acute phase.

### 2.2. Scutellarin Reduces Ischemic Neuronal Apoptosis in the Acute Phase by Targeting Neurotransmitter Activities

In order to explore the molecular mechanism of scutellarin’s neuron protection, a proteomic study was carried out on both the cortex (Figure 2A) and hippocampus (Figure 2E) tissue of rats. Compared with the control group, 480 differentially expressed proteins were identified in the ischemic cortex of the model group, of which 190 proteins were significantly downregulated and 290 proteins were upregulated (Figure 2B). A total of 371 differential proteins were identified between the model group and the scutellarin group, 198 of which were significantly downregulated and 173 were significantly upregulated (Figure 2C). In the hippocampus, a total of 382 differentially expressed proteins were identified between the control group and the model group, of which 199 proteins were significantly downregulated and 183 proteins were upregulated (Figure 2F). Between the model group and the scutellarin group, 217 proteins were significantly downregulated and 188 proteins were significantly upregulated (Figure 2G). A preliminary analysis of biological functions was performed with intergroup differential proteins using IPA software. It was found that scutellarin rapidly regulates various neurotransmitter activities in both the ischemic cortex and hippocampus, of which, glutamatergic pathways are most prominent (Figure 2D,H). In addition, molecular docking was performed to investigate the interaction between the glutamate inhibitor memantine, scutellarin, and glutamate receptors. The docking scores between the memantine and glutamate receptors GRIN1, GRIN2B, and GRM1 were −7.4, −6.0, and −6.4, respectively. Scutellarin showed docking scores of −10.1, −8.4, and −8.7 for GRIN1, GRIN2B, and GRM1, respectively (Table 1). These docking scores were much higher than those for memantine. The amino acid residues Thr126, Gln144, Asp224, and Gln95 of GRIN1 form hydrogen bonds with scutellarin ligands; meanwhile, the amino acid residues of GRIN1, including Leu146, Trp223, Val181, Ser180, Tyr184, Thr94, Trp106, Glu130, Phe92, and Pro124, form hydrophobic interactions with scutellarin ligands (Figure 2I). It was suggested that scutellarin probably interacts with glutamate receptors in a strong and direct way.

Scutellarin’s regulation of the glutamatergic signaling pathway was further investigated using the PRM-MS technique (Table 2). Aberrant expressions of EAAT2, GRIN1, GRIN2, GRM1, CAMKK2, PSD95, and nNOS proteins in the ischemic cortex were found to be significantly redressed upon scutellarin treatment (Figure 3A,B,D–H). Aberrant expressions of EAAT2, GRIN1, CAM and nNOS proteins in the ischemic hippocampus were found to be significantly redressed during scutellarin treatment (Figure 3I–L). Taken together, it was suggested that scutellarin protects neurons from acute ischemic injury through the rapid regulation of glutamatergic signaling.

### 2.3. Changes in the Metabolic Profile of the Ischemic Brain in Acute Phase and Scutellarin Regulation

To investigate the alteration in the metabolic profile of the brains of individuals suffering from acute ischemia, metabolomics was applied to both the cortical and hippocampus tissues of rats. Compared to the control group, 612 different metabolites were identified in the ischemic cortex of the model group, 341 of which were significantly downregulated and 271 were upregulated (Figure 4A). In the hippocampus, a total of 212 differentially expressed metabolites were identified between the control group and the model group, of which 91 metabolites were significantly downregulated and 121 metabolites were upregulated (Figure 4C). In both the cortex and hippocampus, multiple amino acid metabolism pathways were enriched (Figure 4B,D). The results indicated that amino acid metabolism could be a potential target for acute-phase ischemia treatment.

Next, metabolomics was performed on ischemic cortex and hippocampus samples in order to examine the metabolic impact of scutellarin on acute ischemic injury. Further, PCA and OPLS-DA results illustrated aberrant metabolic expression trends that were effectively redressed by scutellarin in both the cortex (Figure 5A–C) and hippocampus (Figure 5D–F). In total, 226 molecules from 2347 ions were found to contribute to good separations between the hippocampus groups (FC > 1.2 or <0.83, *p* < 0.05). In total, 127 molecules from 2396 ions were found to contribute to good separations between the cortical groups (FC > 1.2 or <0.83, *p* < 0.05). The KEGG enrichment results suggested that scutellarin’s metabolic regulation of acute ischemic damage primarily involves a set of amino acid metabolisms in the cortex and hippocampus (Figure 5G,H). The results indicated that scutellarin rapidly regulates amino acid neurotransmissions in the ischemic region of the brain, which is consistent with proteomics results.

Targeted amino acid quantification was performed using the PRM-MS technique and available amino acid standard substances. Upon scutellarin treatment, a total of eight amino acids were found to be significantly regulated in the cortex (Figure 6A–H) and hippocampus (Figure 6I–P), respectively. In the cortex, the regulated amino acids were glutamate, γ-aminobutyric acid, isoleucine, leucine, serine, threonine, phenylalanine, and arginine. In the hippocampus, they were glutamate, γ-aminobutyric acid, valine, serine, threonine, phenylalanine, arginine, and histidine. These results indicated that redressing amino acid levels fast could be a potential strategy for treating acute ischemic stroke.

## 3. Discussion

Cerebral ischemic stroke is a major cause of long-term disability and mortality worldwide. The management of patients’ needs causes a great burden on medical and social care resources. A blocked cerebral blood flow leads to bioenergetic collapse, and the following neurotransmitters communicate that there is disorder in the brain within minutes. Among them, the extracellular release of toxic concentrations of excitatory amino acids, including glutamate and aspartate, is the most discussed [20,21,22]. Given the crucial role of excitatory amino acids in the pathobiology of acute cerebral ischemia, excitatory amino acid neurotransmission, especially glutamate, is considered one of the potential targets for rapid intervention in cerebral ischemic stroke. During the acute phase of cerebral ischemia, most neurons die of excitotoxity, which is induced by the excessive release of glutamate and the subsequent activation of ionotropic receptors [23,24]. Multiple antagonists have been developed in order to reduce histological lesions and provide neuroprotection in animal models; these target glutamate and glycine binding sites, the ion channel site, and the ionotropic glutamate receptor subunit. However, these drugs acting on the glutamatergic system failed to show any advantage in terms of reducing the proportion of deaths, due to safety concerns. These drugs display side effects that include hypertension, hallucinations, nausea, vomiting, and agitation. Worse functional outcomes, or even higher mortality during drug administration, lead to the failure of clinical trials. Immediate toxicity, as well as delayed recovery, remain barriers to the development of drugs for cerebral ischemic stroke.

We decided to explore fast-acting candidates from natural pharmaceutical molecules that have the potential to be used as safer and more effective treatments for acute phase cerebral ischemic stroke. Previous studies have shown that scutellarin has beneficial effects on ischemic lesions in the brain [25], heart [26], and kidneys [27], with the advantage of low toxicity. However, the effect of scutellarin on acute ischemic brain injury remains to be determined. In this study, scutellarin was found to prominently improve nerve function and brain infarction size in MCAO/R model rats within 12 h. Further, histopathological examinations, including H&E and TUNEL staining, were performed on cortical and hippocampus tissue undergoing 12 h reperfusion. The results showed that neuronal damage and apoptosis were significantly alleviated with scutellarin treatment. Molecularly, upregulated cleaved caspase 3 in the model group was reduced with scutellarin treatment. Therefore, scutellarin has been proven to be beneficial and fast-acting in the treatment of acute cerebral ischemic injury in an animal model.

With proven efficacy, further investigations regarding the regulatory effects of scutellarin were conducted. Analyses were performed on cortex and hippocampus tissue, respectively, to enhance knowledge of the pathogenesis and therapeutic targets of acute cerebral ischemia. With proteomics studies, scutellarin was found to regulate multiple neurotransmitters activities, with glutamatergic system regulation scoring highest. Scutellarin could significantly inhibit glutamatergic signaling. Based on the PRM-MS results, aberrant levels of glutamate receptors, including EAAT2, GRIN1, GRIN2B, and GRM1, as well as other glutamatergic pathway-involving proteins, including CAMKK2, PSD95, and nNOS, were found to be significantly redressed in the ischemic cortex. In the hippocampus, EAAT2, GRIN1, nNOS, and CAM were significantly regulated. PSD95 protein is involved in the functional interaction between the GRIN2B subunit and nNOS. It is a promising target for the blockage of NMDA receptor activation blockage [28]. CaMKII is a critical mediator of the pathological glutamate signaling that is related to neural excitotoxicity [29]. It could be phosphorylated and activated under the stimulation of CAMKK or CaM. EAAT2 is a transporter that prevents excitotoxicity by removing glutamate from the extracellular space of synapses.

Moreover, metabolomics were performed on both the cortex and hippocampus. Consistent with proteomics findings, amino acid metabolism and neurotransmission play roles in scutellarin treatment of acute ischemic injury. The glutamatergic pathway was found to score highest in terms of KEGG pathway enrichment. Amino acids are essential for proper neurotransmission by functioning as precursors to neurotransmitters and exerting multiple neuromodulatory effects. Several amino acids even exert neurotransmitter-like effects in the brain. The disorder of amino acid metabolism would lead to deficiencies in neurotransmission. With PRM-MS, amino acids in the ischemic region were absolutely quantified based on the standard substances available. Amino acids, including glutamate, γ-aminobutyric acid, isoleucine, leucine, valine, serine, threonine, phenylalanine, arginine, and histidine, were found to be significantly redressed by scutellarin. In addition to the excitatory amino acid glutamate, and inhibitory amino acid γ-aminobutyric acid, other amino acids also have an impact on neurotransmission. As branched-chain amino acids (BCAAs), isoleucine, leucine, and valine modulate the synthesis and release of aromatic amino acids by competing with them in the neural system [30,31,32,33,34]. In addition, BCAAs also convert into glutamate to enhance excitotoxicity [35,36,37,38]. Serine participates in neurotransmission by aiding the release of glutamate and aspartate [39]. Both serine [40,41,42,43] and threonine [44,45] serve as precursors of glycine synthesis. Phenylalanine and histidine act as precursors to catecholamines, serotonin, and histamine synthesis [33,41,46,47]. Arginine always serves as a precursor for NOx [48], which is critical for protecting against brain injury.

Taken together, scutellarin could greatly improve ischemic brain damage in the acute phase by interacting with glutamate receptors, regulating not only the glutamatergic signaling that is implicated proteins, but also multiple amino acid levels in both the ischemic cortex and hippocampus. This could reduce neuronal apoptosis caused by excitatory amino acid toxicity. This study provides evidence for scutellarin’s fast-acting effect on acute ischemic brain injury, and enhances knowledge about the early therapeutic targets for cerebral ischemic stroke.

## 4. Materials and Methods

### 4.1. Chemicals and Reagents

Scutellarin standard substances (content ≥98%) were purchased from Chengdu Herbpurify Co., Ltd., Chengdu, China. Methanol, formic acid and acetonitrile were purchased from Fishier Scientific, Fairlawn, OH, USA. Trypsin (modified, sequencing grade) was purchased from Promega, Madison, WI, USA. The other chemicals were purchased from Sigma-Aldrich (St Louis, MO, USA) unless stated otherwise.

Control group: at 0 h, 1.5 h and 7.5 h after MCAO operation modeling, normal saline administration was performed via tail vein injection on rats without modeling.

Model group: MCAO/R model was used in the model group. For MCAO surgery, the rats were continuously anesthetized with isoflurane (1.5% of isoflurane/air (*v*/*v*)). Right MCAO (Middle Cerebral Artery Occlusion) was performed with a 4-0 silk suture. Reperfusion was performed 1.5 h later, followed by normal saline administration via tail vein injection at 0 h, 1.5 h and 7.5 h after MCAO operation modeling.

### 4.2. Animals and Grouping

Sprague Dawley (SD) rats (150 ± 10 g) (Si Pei Fu Laboratory Animal Co. Ltd., Beijing, China, SCXK (Jing) 2019-0010) were used in this experiment. The experiments were performed in accordance with the Guide for the Care and Use of Laboratory Animals of Beijing University of Chinese Medicine, with protocol approved by the Committee on Research Practice of Beijing University of Chinese Medicine (Approval ID: BUCM-4-2020093013-3163, Approval date: 30 September 2020). After being housed in polypropylene cages (5/cage) with free access to food and water (12 h light–dark cycle, temperature of 24 ± 2 °C, relative humidity of 55% ± 5%) for 1 week, rats were randomly divided into control, model, scutellarin high-dose, and scutellarin low-dose groups (12/group). Scutellarin was dissolved in DMSO with a concentration of 216.28 mM and was then diluted with normal saline for injection.

Scutellarin high-dose group: at 0 h, 1.5 h and 7.5 h after MCAO operation modeling, scutellarin administration (12 mg/kg body weigh) was performed via tail vein injection on rats that were subjected to MCAO/R.

Scutellarin low-dose group: at 0 h, 1.5 h and 7.5 h after MCAO operation modeling, scutellarin administration (6 mg/kg body weight) was performed via tail vein injection on rats that were subjected to MCAO/R.

All specimens were obtained at 12 h after reperfusion for further research.

### 4.3. TTC and TUNEL Staining

After treatment of MCAO/R model rats with vehicle or scutellarin, the brains were cut coronally into five sections with an average length, then stored in 2% TTC at 37 °C for 30 min and 4% buffered formalin phosphate for 30 min. Scion image (Alpha 4.0.3.2, Scion Corporation, Frederick, MD, USA) and Image Pro-Plus 6.0 were applied for image analysis with semi-automatic image segmentation. TUNEL staining was performed following the protocol, which was provided by the manufacturer.

### 4.4. Protein Sample Preparation and NanoLC-Orbitrap Fusion Lumos MS

The right-hemisphere brain samples of three biological replications of each group were centrifuged at 16,000× *g* for 30 min at 4 °C after homogenization and lysis. The supernatant was collected and stored in aliquots at 20 °C. Following this, all the samples were digested with trypsin and processed using the FASP method, which was previously described by Wisniewski [49].

### 4.5. Protein Identification and Quantification

The MS/MS spectra data were searched against the Uniprot KB/SwissProt Rat database of canonical sequences using automatic decoy database search engines via Proteome Discoverer 2.4 (Thermo Fisher Scientific, San Jose, CA, USA). The quantitative value of each identified protein was determined by unique peptides only. Bias correction for unequal mixes in the different labeled samples was performed. All of the data were normalized by bias correction and normal distribution evaluation using scatter plots; R. Student’s *t*-test (two-tail) was performed between groups.

### 4.6. Metabolite Sample Pretreatment and UHPLC-Q-Exactive Orbitrap MS Analysis

Brain tissue was collected into tubes containing methanol/acetonitrile solution (3:1, *v*/*v*) and centrifuged for 14,000× *g*, 15 min at 4 °C after vortex mixing. The supernatant was obtained and sample preparation was carried out using protein precipitation with 50% acetonitrile. To validate the analytical methodology, pooled quality control (QC) samples were prepared by mixing samples from each group. Five QC samples were analyzed before sample sequencing, and during the analysis of the sample sequence, one QC sample was run after every five injections [50].

Metabolomics analysis was performed on a Thermo Scientific Vanquish UHPLC coupled to a Q Exactive Orbitrap MS system, which was equipped with an electrospray ionization source. The operating parameters were as follows: spray voltage, 3.5 KV; sheath gas pressure, 35 arb; auxiliary gas pressure, 10 arb; capillary temp, 300 °C; ion source temp, 350 °C; scan modes, MS (Full Scan, *m*/*z* 100–1200) and data-dependent acquisition MS^2^ (resolution of 17,500, normalized collision energy of 35 eV, stepped normalized collision energy of 30 and 40 eV); and scan range, *m*/*z* 80–1200. Taking polar and minor polar components into consideration, an ACQUITY UHPLC BEH C18 column (1.7 um 2.1 mm × 100 mm) and ACQUITY UHPLC HILIC column (1.7 um 2.1 mm × 100 mm) were equipped for complete data collection and analyses. The mobile phases were combined with 0.1% formic acid in water (A) and 0.1% formic acid in acetonitrile (B).

UHPLC gradient conditions: 0–10 min, 8–30% (A); 10–12 min, 30–30% (A); 12–13 min, 30–8% (A); and 13–15 min, 8–8% (A).

For metabolomics study, the raw data were imported into Sieve 2.1 software (Thermo Fisher Scientific Inc., San Jose, CA, USA) for pretreatment. Then, a Principal Components Analysis (PCA) and Orthogonal Partial Least Squares-Discriminant Analysis (OPLS-DA) were carried out on the data using SIMCA-P13.0 software (Umetrics AB, Umea, Sweden). Then, identification of the differential metabolites was performed, based on Compound Discoverer 2.0, the Metlin, and HMDB databases, as well as manual matching.

### 4.7. Bioinformatic Data Analysis and Molecular Docking

The obtained differential metabolites were analyzed through an MeV (Multi Experiment Viewer, v4.8, TIGR) for hierarchical cluster analysis and K-mean cluster analysis. All experimental data were expressed as means ± SD, and the statistical significance was calculated using Student’s *t* test in the SPSS (version 22.0); *p*-values lower than 0.05 were considered significant. Principal Components Analysis (PCA) and Orthogonal Partial Least Squares-Discriminant Analysis (OPLS-DA) were performed with SIMCA-P 13.0 software (Umetrics AB, Umea, Sweden). The IPA software and database KEGG were applied for the pathway enrichments.

The molecular docking studies were performed on a Glide program incorporated in the AutoDock Vina software, based on the structure documents of GRIN1 (PDBID:5KCJ), GRIN2B (PDBID:3JPW), and GRM1 (PDBID:1EWV).

### 4.8. Parallel Reaction Monitoring-Mass Spectrometry (PRM-MS)

According to the quantitative proteomics result, characteristic peptides of target proteins were selected for PRM validation. An Orbitrap Fusion Lumos Tribrid mass spectrometer coupled with an EASY-nLC 1200 system (Thermo Scientific, New York, NY, USA) was employed for PRM study. Raw data were processed in Skyline (version 19.1, MacCoss Laboratory, University of Washington, Washington, WA, USA) to generate extracted-ion chromatograms and peak integration. The distribution of the relative intensities of multiple transitions, which is generated from the same precursor ion, should be in accordance with the theoretical distribution. The peak area of each characteristic peptide was obtained with Skyline analysis based on >4 sub-ions, which are of high abundance and are as consistent as possible in terms of their retention time in the secondary mass spectrometry.

### 4.9. Western Blotting Assay

Protein was obtained by applying RIPA Lysis Buffer (Beyotime, P0013B), protease and phosphatase inhibitors (Applygen, All-in-One, 100×) to cells or tissue samples. Membranes were incubated with cleaved caspase3 (1:1000, Cell Signaling Technology, 9664) and beta-actin (1:2000, Proteintech, 20536-1-AP) overnight. Images were developed by a chemiluminescence imager after the incubation of horseradish peroxidase-labeled secondary antibodies.

## 5. Conclusions

In this study, scutellarin was found to exert quick regulations on neurotransmitter activities, especially in the glutamate-related pathway. It protects against ischemic brain damage in the acute phase by regulating the activity of neurotransmitters and reducing the toxicity of excitatory amino acids in neurons.

## Figures and Tables

**Figure 1 molecules-28-03181-f001:**
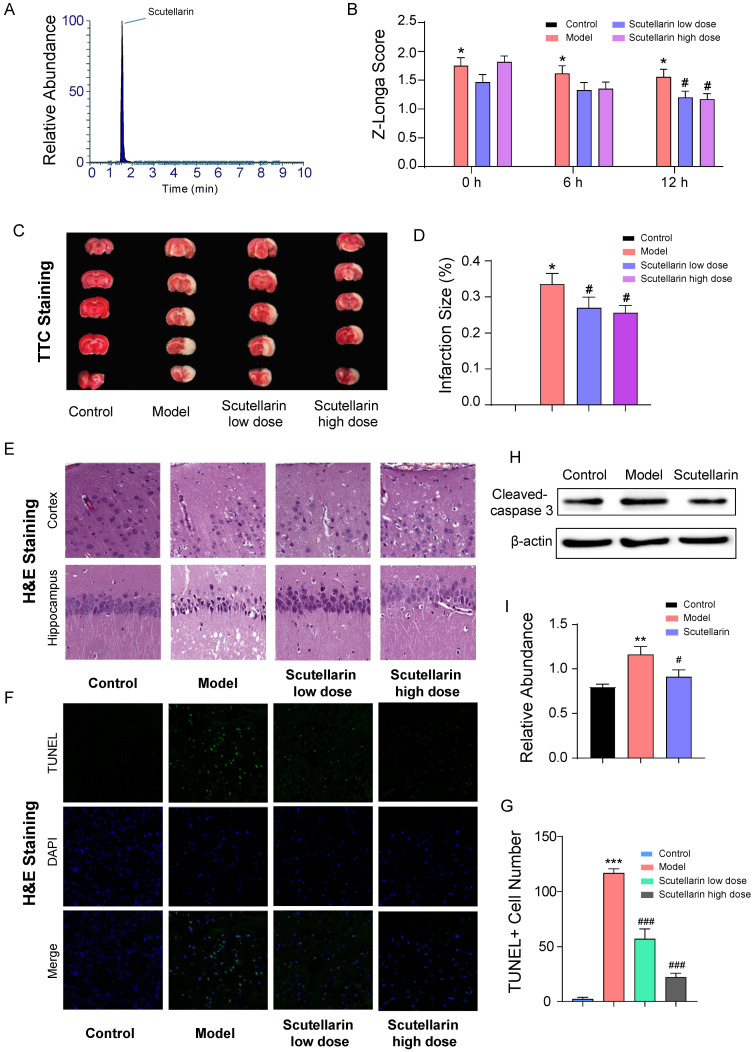
Protective effect of Scutellarin against acute ischemic injury in MCAO/R model rats. (**A**) The scutellarin used in this study is 98% pure. (**B**) Rat Z-Longa scores within 12 h after reperfusion. (**C**,**D**) TTC staining images (**C**) and infarction size (**D**) of rat brains at 12 h after reperfusion. (**E**) Scutellarin administration alleviates neuronal damage in both the cortex and hippocampus. (**F**,**G**) TUNEL staining illustrates that apoptotic cell numbers are significantly reduced upon scutellarin treatment. (**H**,**I**) The level of cleaved-caspase 3 in rats’ brain was down-regulated upon scutellarin treatment. Data are presented as mean ± *SD*, *t*-test, vs. control group: * *p* < 0.05, ** *p* < 0.01, *** *p* < 0.001 vs. model group: # *p* < 0.05, ### *p* < 0.001.

**Figure 2 molecules-28-03181-f002:**
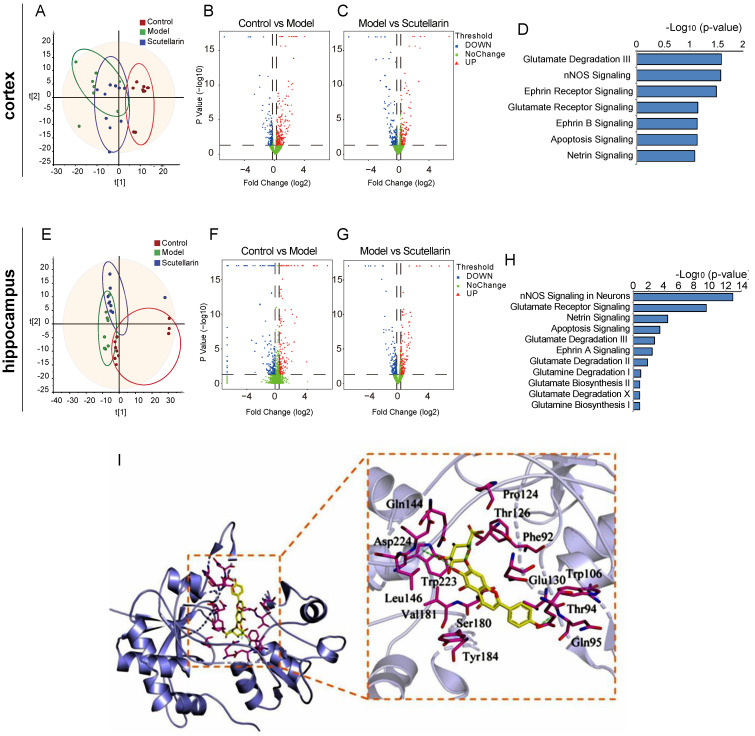
Scutellarin’s regulation of biological processes in the brains of MCAO/R model rats at 12 h after reperfusion. (**A**,**E**) PCA results of proteomics data showing the redressing effect of scutellarin on aberrant protein expression in the cortex (**A**) and hippocampus (**E**), which are induced by MCAO/R modeling. (**B**,**C**,**F**,**G**) Volcano plots showing the differential proteins between the groups that were screened for further biological pathway analysis. (**D**,**H**) IPA biological pathway enrichment results of cortex proteomics (**D**) and hippocampus proteomics (**H**). (**I**) Molecular docking results showing the interaction between scutellarin and GRIN1.

**Figure 3 molecules-28-03181-f003:**
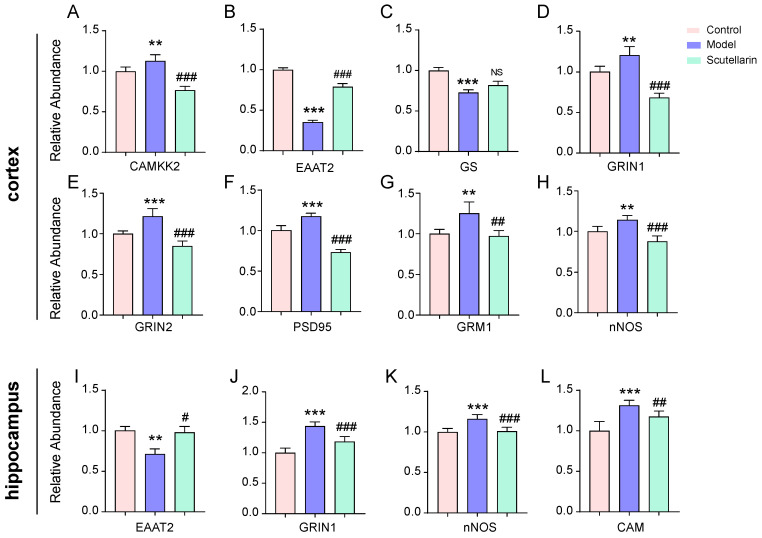
Targeted quantification of glutamatergic signaling-related proteins in the brains of MCAO/R model rats at 12 h after reperfusion. Scutellarin exerts significant reparations on CAMKK2 (**A**), EAAT2 (**B**), GRIN1 (**D**), GRIN2 (**E**), PSD95 (**F**), GRM1 (**G**), and nNOS (**H**) in cortex, but its effect on GS (**C**) is not statistically significant. Scutellarin exerts significant reparations on EAAT2 (**I**), GRIN1 (**J**), Nnos (**K**) and CAM (**L**) in the ischemic hippocampus. Data are presented as mean ± *SD*, *t*-test vs. control group: ** *p* < 0.01, *** *p* < 0.001; vs. model group: # *p* < 0.05, ## *p* < 0.01, ### *p* < 0.001.

**Figure 4 molecules-28-03181-f004:**
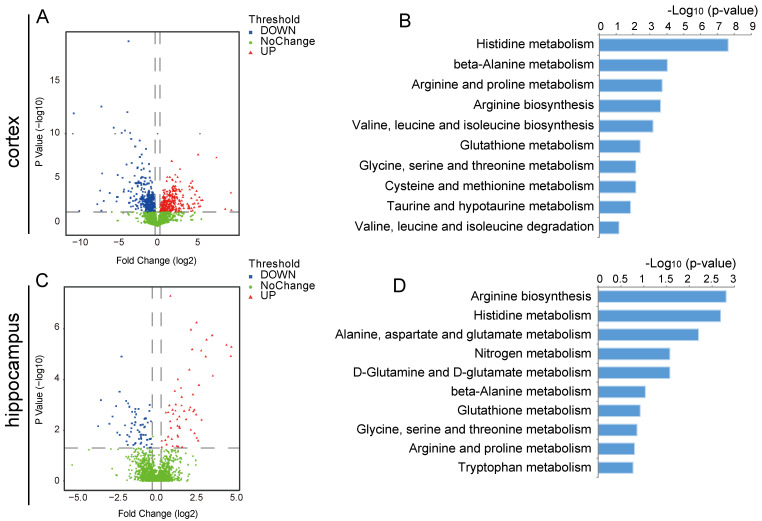
Metabolic characteristics in the acute ischemic cortex and hippocampus. (**A**,**C**) Volcano plots showing protein differences between control and model groups in cortex (**A**) and hippocampus (**C**). (**B**,**D**) KEGG metabolic pathway enrichment results of cortex (**B**) and hippocampus (**D**).

**Figure 5 molecules-28-03181-f005:**
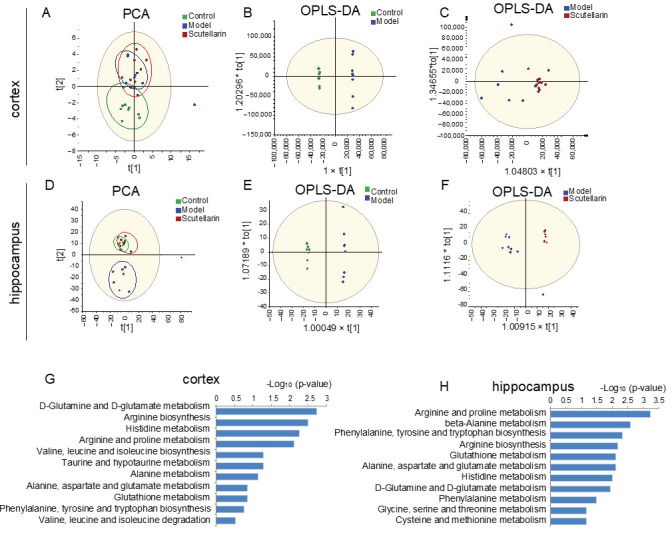
Scutellarin’s regulation of metabolic activities in the brains of MCAO/R model rats at 12 h after reperfusion. (**A**,**D**) The PCA results of metabolomics data show the redressing effect of scutellarin on the expression of aberrant metabolites in the cortex (**A**) and hippocampus (**D**), induced by MCAO/R modeling. (**B**,**C**,**E**,**F**) OPLS-DA on LC-MS data of common metabolites in cortex and hippocampus of control group, model group, and scutellarin group. (**B**) Control vs. model groups for cortex. (**C**) Control vs. scutellarin groups for cortex. (**E**) Control vs. model groups for hippocampus. (**F**) Control vs. scutellarin groups for hippocampus. (**G**,**H**) The KEGG metabolic pathway enrichment results for cortex metabolomics (**G**) and hippocampus metabolomics (**H**).

**Figure 6 molecules-28-03181-f006:**
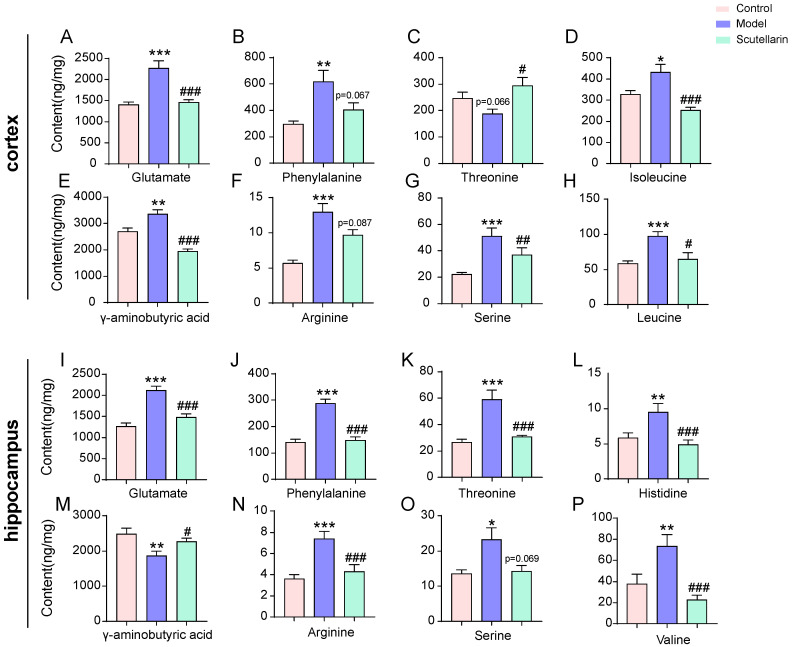
Targeted quantification of amino acids in cortex (**A**–**H**) and hippocampus (**I**–**P**) of MCAO/R model rats at 12 h after reperfusion. Data are presented as mean ± *SD*, *t*-test, vs. control group: * *p* < 0.05, ** *p* < 0.01, *** *p* < 0.001; vs. model group: # *p* < 0.05, ## *p* < 0.01, ### *p* < 0.001.

**Table 1 molecules-28-03181-t001:** The docking score between memantine, scutellarin and glutamate receptors.

Protein	Grid Size	Docking Score (kcal/mol)	
		Memantine	Scutellarin
GRIN1 GRIN2 GRM1	40 × 40 × 40 40 × 40 × 40 40 × 40 × 40	−7.4 −6.0 −6.4	−10.1 −8.4 −8.7

**Table 2 molecules-28-03181-t002:** The characteristic peptides of targeted proteins.

Protein	Protein Accession	Unipeptide Sequence Applied
EAAT2	Peptide 1	KNDEVSSLDAFLDLIR
	Peptide 2	SELDTIDSQHR
	Peptide 3	MHEDIEMTK
	Peptide 4	SADC[+57]SVEEEPWKR
GS	Peptide 1	DIVEAHYR
	Peptide 2	RPSANC[+57]DPYAVTEAIVR
	Peptide 3	TC[+57]LLNETGDEPFQYK
GLS	Peptide 1	VADYIPQLAK
	Peptide 2	FSPDLWGVSVC[+57]TVDGQR
	Peptide 3	MAGNEYVGFSNATFQSER
Grin1	Peptide 1	VEFNEDGDRK
	Peptide 2	IIWPGGETEKPR
	Peptide 3	YADGVTGR
Grin2b	Peptide 1	IISENKTDEEPGYIK
	Peptide 2	NLTNVDWEDR
Grm1	Peptide 1	GEVSC[+57]C[+57]WIC[+57]TAC[+57]K
	Peptide 2	KPGAGNANSNGKSVSWSEPGGR
Grik5	Peptide 1	ETLSVRMLDDSRDPTPLLK
	Peptide 2	MVEYDGLTGR
Gria2	Peptide 1	GADQEYSAFR
	Peptide 2	ADIAIAPLTITLVR
Gria3	Peptide 1	INTILEQVVILGK
	Peptide 2	YTSALTHDAILVIAEAFR
	Peptide 3	ADIAVAPLTITLVR
CaM	Peptide 1	EAFSLFDKDGDGTITTK
	Peptide 2	ELGTVMR
	Peptide 3	MKDTDSEEEIR
Camkk1	Peptide 1	MERSPAVC[+57]C[+57]QDPRAELVER
	Peptide 2	SPAVC[+57]C[+57]QDPR
	Peptide 3	HGEEPLPSEEEHC[+57]SVVEVTEEEVK
Camkk2	Peptide 1	GPIEQVYQEIAILK
	Peptide 2	IFSGKALDVWAMGVTLYC[+57]FVFGQC[+57]PFM[+16]DER
nNOS	Peptide 1	MDLDM[+16]RK
	Peptide 2	LSEEDAGVFISR
	Peptide 3	SQAYAKTLC[+57]EIFK
	Peptide 4	FSVFGLGSR
PSD95	Peptide 1	GNSGLGFSIAGGTDNPHIGDDPSIFITK
	Peptide 2	FGDVLHVIDAGDEEWWQAR
	Peptide 3	ANDDLLSEFPDK
	Peptide 4	FGSC[+57]VPHTTRPK
PLA2	Peptide 1	DPRYGASPLHWAK
	Peptide 2	QPAELHLFRNYDAPEAVR

## Data Availability

The data that support the findings of this study are available from the corresponding author X.L., Yan, upon reasonable request.

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
