# Peer review of "Scutellarin Alleviates Ischemic Brain Injury in the Acute Phase by Affecting the Activity of Neurotransmitters in Neurons"

_molecules, 2023, doi:10.3390/molecules28073181_

Round 1

Reviewer 1 Report

Title: Scutellarin alleviates ischemic injury of brain in acute phase by 2 targeting neurotransmitters activity in neuron

The present study has investigated whether scutellarin is a promising molecule for the acute phase of ischemic brain damage. The study has been well performed and also presented expertly. For inducing the experimental model, middle cerebral artery occlusion and reperfusion modeling were used. TTC, H&E, TUNEL staining, and western blotting methods were used to explain the neuroprotective effect of scutellarin. Proteomics and metabolomics studies were added to explore the neuroprotective mechanisms of scutellarin. A molecular docking study also supported the interaction of scutellarin molecules on glutamate receptors.  The study can be accepted after following a few minor modifications.

1.       Line 45, needs to read as “is caused”.

2.       Line 60, “brain damage is still weak and need further clarify” needs to be modified.

3.       Line 75, needs to read as “which is suggesting”.

4.       Lines 273-275, “Scutellarin……Promega)” needs to be modified as complete sentences.

5.       Line 295, scutellarin was administrated by tail vein injection. Detailed information about the injection should be added here. Is the compound freely soluble in water?, how it was dissolved and injected through the tail vein?

6.       The conclusion of the study is missing in the manuscript. 

Reviewer 2 Report

Dear Editor,

It is my privilege to review the article titled “Scutellarin alleviates ischemic injury of brain in acute phase by targeting neurotransmitters activity in neuron” submitted for publication in Molecules by Wang et al.

My view is that this article has a great potential but requires too much work to be considered for publication at this stage. It is a challenge to read it because of suboptimal flow and the approximate use of the English language.

Details are provided below:

Title

It should be rephrased for clarity, for example,

“Scutellarin alleviates ischemic brain injury in the acute phase by affecting the activity of neurotransmitters in neurons”.

Abstract

a)       The section is very difficult to read. It is full of grammatical errors and the flow is not good.

b)      The section is merely descriptive, with no actual data.

c)       The conclusion is not supported by the description of the section.

d)      A rewriting is required, with emphasis on language and flow

Introduction

a)       Add references to all major claims.

b)      The flow is very poor, making the comprehension extremely difficult.

c)       "Scutellarin had been found to be beneficial to ischemic diseases such as stroke and myocardial infarction by fighting against inflammation, oxidation, and coagulation in previous studies. It shows high efficacy and low toxicity in clinical application.". From this passage, and since " Cerebral ischemic stroke ac-38 counts for 60%-70% of the total strokes seen in patients" according to the authors, this work has little novelty. The authors could present their case differently to highlight the actual data gaps of the field and clearly state what their goal is.

d)      " therapeutic window" is used for a drug not for a disease. Word choice is poor in general. This is just an example

e)      The study aim is not clearly provided.

f)        A rewriting is required, with emphasis on language and flow.

Results

a)       The authors have well-designed tables and graphs, but the section should be rewritten because of its poor description. Please extract numbers and add to the text. The reader must not refer to the tables or figures to have a good idea of what’s going on.

b)      A rewriting is required, with emphasis on language and data reporting.

Discussion

This part should also be rewritten. It is merely background, and the data are not appropriately discussed. While rewriting, excessive background should be avoided, and the data should be presented in light of currently available reports. In addition, pay attention to the language.

Materials and Methods

All materials should include their manufacturers and countries of manufacture.

Also pay attention to the language and clarity.
